# Residue Depletion Profile and Estimation of Withdrawal Period for Sulfadimethoxine and Ormetoprim in Edible Tissues of Nile Tilapia (*Oreochromis* sp.) on Medicated Feed

**DOI:** 10.3390/ani13152499

**Published:** 2023-08-03

**Authors:** Lucas Victor Pereira de Freitas, Carlos Augusto Alvarenga da Mota Júnior, Marina Alves Damaceno, Juliana Grell Fernandes Silveira, Ana Carolina Vellosa Portela, Sarah Chagas Campanharo, Agnaldo Fernando Baldo da Silva, Inácio Mateus Assane, Fabiana Pilarski, James Jacob Sasanya, Jonas Augusto Rizzato Paschoal

**Affiliations:** 1Department of Biomolecular Sciences, School of Pharmaceutical Sciences of Ribeirão Preto, University of São Paulo (USP), Ribeirao Preto 14040-900, SP, Brazil; 2Laboratory of Microbiology and Parasitology of Aquatic Organisms, Aquaculture Center of Unesp, Sao Paulo State University (Unesp), Jaboticabal 14884-900, SP, Brazil; 3International Atomic Energy Agency, A-1400 Vienna, Austria

**Keywords:** aquaculture, tilapia, food safety, veterinary drug residues, withdrawal period, QuEChERS, LC-MS/MS

## Abstract

**Simple Summary:**

The use of veterinary drugs in fish farms is required to maintain animal health, minimize economic losses, and increase productivity. This study investigated the residue depletion profile of sulfadimethoxine (SDM) and ormetoprim (OMP) in Nile tilapia (*Oreochromis* sp.) after oral administration aiming to estimate a minimum withdrawal. The medicated feed used was prepared and analyzed to ensure suitability for animal treatment and prevent environmental contamination. After treatment, fish were euthanized at different times, their fillets were collected, and the drug residues were determined. Two analytical methods were developed to quantify SDM and OMP in feed and fillet samples. Meanwhile, a withdrawal period of 252 °C days was estimated.

**Abstract:**

Sulfadimethoxine (SDM) and ormetoprim (OMP) are antimicrobials used in combination to treat bacterial infections in fish farming. The use of this drug combination is not yet regulated in some countries, such as Brazil. Due to the lack of regulated drugs for aquaculture in Brazil, this study investigated the residue depletion profile of SDM and OMP in Nile tilapia (*Oreochromis* sp.) after oral administration. Fish were treated with medicated feed containing a 5:1 ratio of SDM:OMP at the dose of 50 mg kg BW^−1^ for five consecutive days with an average water temperature of 28 °C. The drugs were incorporated into the feed by using a gelatin coating process which promoted homogeneity in drug concentration and prevented the drug leaching into the water during medication. The SDM and OMP determination in fish fillets (muscle plus skin in natural proportions) was performed using the QuEChERS approach followed by LC-MS/MS quantification. The analytical method was validated according to Brazilian and selected international guidelines. A withdrawal period of 9 days (or 252 °C days) was estimated for the sum of SDM and OMP residues at concentration levels below the maximum residue level of 100 µg kg^−1^.

## 1. Introduction

Aquaculture, one of the fastest-growing food sectors in the world, has been increasingly recognized for its contribution to global food security and nutrition in the twenty-first century [1,2]. In 2020, the global capture fishery production was 90.3 million tons, with Nile tilapia (*Oreochromis* sp.) production close to 7 million tons in 124 countries and valued at USD 12.342 billion [3,4]. However, bacterial infection is a significant challenge to fish farming systems, causing substantial economic losses. Thus, antimicrobials are commonly used in aquaculture to treat animal diseases and mitigate losses [5,6].

Unlike other food production sectors, aquaculture, especially in countries where this sector is developing, still lacks regulated drugs which may lead to a misuse of various veterinary drugs including banned substances [7,8]. These practices lead to failures in treatment, environmental contamination, emergence of bacterial resistance, and other risks to human health. Studies aim at understanding the use of veterinary drugs in aquaculture given the wide-ranging effects on consumers, therapeutics, and the environment among others. Such studies are also relevant to the One Health concept attempting to address responses and actions at the animal–human–ecosystem interfaces [9].

The most used antimicrobials in aquaculture worldwide are amoxicillin, enrofloxacin, erythromycin, tetracycline, florfenicol, flumequine, sarafloxacin, sulfadimethoxine, ormetoprim, and oxytetracycline [10,11]. Sulfonamides are commonly used in farmed fish, including combinations with trimethoprim and ormetoprim [5].

Sulfadimethoxine (SDM) is a relatively potent and long-acting sulfonamide that is effective when used alone or in combination with ormetoprim (OMP) in treating bacterial infections [12,13]. Rapid absorption, short half-life, increase in efficacy as well as lower incidence of bacterial resistance are some of the attributes of these drugs in combination [5,14]. A good example is Romet 30^®^, an antibacterial drug approved in the United States, Canada, and other countries which contains SDM and OMP in a ratio of 5:1 and is recommended for applications such as the control of furunculosis in freshwater-reared salmonids (trout and salmon) caused by *Aeromonas salmonicida*, and the control of bacterial infections in catfish caused by *Edwardsiella ictaluri* (enteric septicemia of catfish) [15,16]. The drug combination is also used in an extra-label manner to treat other diseases in a variety of fish species [17].

Despite the effectiveness and broad spectrum of action of the combination of SDM and OMP, the use of these antimicrobials in fish farming is not regulated in many countries, such as Brazil, which currently has only two antimicrobials (florfenicol and oxytetracycline) regulated for use in this system [18]. Although SDM and OMP help in reducing the incidence of infectious diseases, their use may lead to residues in food products, posing potential health risks to consumers such as allergic reactions, carcinogenesis, or induction of the development of pathogenic bacteria-resistant strains [5,19,20]. Thus, the Maximum Residue Limits (MRLs) in foodstuffs of animal origin have been established for these and other veterinary drug residues [19,21]. The MRL adopted for SDM and OMP in fish fillets (muscle plus skin in natural proportions) is 100 µg kg^−1^ [22,23,24,25]. Nevertheless, studies of residue depletion and establishing the withdrawal times for edible tissue concentrations to below MRL are crucial to safe veterinary drug use for consumer and environmental protection.

Therefore, this study aimed to investigate the residue depletion profile of SDM and OMP and estimate the withdrawal period in Nile tilapia (*Oreochromis* sp.) on medicated feed and cultured under Brazilian climatic conditions. Incorporation of these drugs in the feed was evaluated to ensure a homogeneous drug concentration among the pellets with a low risk of drug leaching into the surrounding water during treatment. Two analytical methods for the determination of SDM and OMP in feed and Nile tilapia fillets were developed and validated according to Brazilian and targeted international guidelines like European Community and Codex Alimentarius.

## 2. Materials and Methods

### 2.1. Chemicals and Reagents

The analytical standards for OMP and SDM (purity ≥ 99.0% *w*/*w*) were purchased from Dr. Ehrenstorfer (Augsburg, Germany) and Sigma-Aldrich (St. Louis, MO, USA), respectively. Sulfamethoxazole with purity ≥ 99.0% (*w*/*w*) obtained from Sigma-Aldrich (St. Louis, MO, USA) was used as an internal standard (IS).

HPLC-grade methanol (MeOH) and acetonitrile (ACN) were obtained from J. T. Baker (Xalostoc, Mexico). Acetic acid was purchased from Merck (Readington, NJ, USA). Anhydrous magnesium sulfate (MgSO_4_—from Sigma-Aldrich, São Paulo, Brazil) and sodium chloride (NaCl—from Sigma-Aldrich, São Paulo, Brazil) were used in sample preparation. Primary secondary amine (PSA) was obtained from Agilent Technologies (Santa Clara, CA, USA). Benzocaine with a purity higher than 98% (*w*/*w*) from Sigma-Aldrich, São Paulo, Brazil) was employed to prepare fish for slaughter. Ultrapure water (18.2 MΩ) was obtained from a Milli-Q 3-UV System from Merck Millipore (Molsheim, France).

### 2.2. Equipment 

Analytical standards, samples, and reagents were weighed using analytical balances from Shimadzu (Kyoto, Japan) and Marte Científica (São Paulo, Brazil). The following types of equipment were used for the sample preparation procedures: food processor from Walita (São Paulo, Brazil), vortex AP56 model (Phoenix Luferco, Araraquara, Brazil), ultrasonic cleaner Maxiclean USC-1450 model (Unique, São Paulo, Brazil), and refrigerated centrifuge Himac CF5RX model (Hitashi, Chiyoda, Japan). Polytetrafluoroethylene filtration membranes (0.22 µm pore size) from Sartorius Stedim Biotech (Göttingen, Germany) were used to filter the sample extracts before chromatographic analysis.

An LC system (Shimadzu, Kyoto, Japan) equipped with an autosampler, binary pumping, and thermostatic column compartment was used for the chromatographic analyses. The LC was interfaced to a triple-quadrupole mass spectrometer Quattro LC (Micromass Limited, Altrincham, UK) by an electrospray ionization source (ESI). Data processing and instrument control were performed by the Mass-Lynx 4.0 software.

### 2.3. Standard Solutions

The standard stock solutions of OMP, SDM, and the IS at the concentration of 1 mg mL^−1^ were prepared by dissolving the analytical standards in MeOH and kept at −20 °C. Working solutions ranging from 0.2 µg mL^−1^ to 5 µg mL^−1^ were prepared daily through the dissolution of the stock solutions in H_2_O/MeOH 55/45 (*v*/*v*), both containing 0.5% formic acid and used immediately after preparation.

### 2.4. Analytical Method for SDM and OMP Determination in Feed

The fish feed was ground with a mortar and pestle and 0.250 g portions weighed in a 15 mL polypropylene tube to which 5.0 mL of MeOH was then added. The mixture was vortexed for 30 s and placed in an ultrasonic bath for 10 min and centrifuged at 2900× *g* for 5 min. An aliquot of 100 μL of the supernatant was collected, then transferred to a 10 mL volumetric flask and made up to the mark with H_2_O/MeOH 55/45 (*v*/*v*), both containing 0.5% formic acid. An aliquot (2 mL) of the extract was filtered into a vial through a 0.22 μm syringe filter and 5 μL injected into the LC-MS/MS system. 

Chromatographic separations were performed using a reversed-phase analytical column X-Terra MS C18 column (3.9 mm × 100 mm × 3.5 μm, Waters, Milford, MA, USA) and guard column with the same filling (3.9 mm × 20 mm, 3.5 µm, Waters, Milford, MA, USA). The column temperature was set at 25 °C. Ultrapure water (solvent A) and methanol (solvent B), both containing 0.5% formic acid, were used to compose the mobile phase running in the isocratic mode (55% A and 45% B) at a flow rate of 0.4 mL min^−1^. The LC-MS/MS interface operated in positive mode (ESI+). The source block and the desolvation gas temperatures were set at 100 °C and 400 °C, respectively. Nitrogen was used both as nebulization and drying gas, set to operate at flow rates of approximately 75 and 470 L h^−1^, respectively. Argon was used as collision gas at 1.78 × 10^−3^ mbar. The analyses were performed in the selected reaction monitoring mode (SRM) to monitor two ion transitions for each analyte. The detailed SRM acquisition parameters are presented in Appendix A.

The reliability of the analytical method to quantify OMP and SDM in fish feed was assessed according to the validation guide proposed by the Brazilian Ministry of Agriculture for methods intended for the determination of drugs in animal feed and other products [26]. The evaluated parameters were selectivity, matrix effect, linearity, accuracy, precision, and robustness. Statistical tests were performed considering the significance level of 0.05.

### 2.5. Analytical Method for SDM and OMP Determination in Fish Fillet

The sample preparation procedure based on modified QuEChERS was used to extract the target analytes from Nile tilapia fish fillet samples (muscle plus skin in natural proportions). The samples were ground in an analytical mill, 1 g was transferred to a 50 mL polypropylene tube, 10 mL of the extractor phase composed of ACN/H_2_0 (8/2, *v*/*v*) was added to the sample and the mixture was vortexed for 30 s and placed in an ultrasonic bath for 15 min. Anhydrous MgSO_4_ (2.0 g) and 1.0 g of NaCl were added, and the mixture was shaken for 1 min followed by centrifugation at 2900× *g* for 6 min at 4 °C. The supernatant (3 mL) was transferred to a 15.0 mL polypropylene tube containing 900 mg of MgSO_4_ and 150 mg of PSA, shaken for 0.5 min, and centrifuged at 2900× *g* for 6 min at 4 °C. The extract was filtered into a vial through a syringe filter, and a volume of 10 μL was injected into the LC-MS/MS system for analysis using the same parameters as in the analytical method for feed analysis above.

The International Cooperation on Harmonization of Technical Requirements for Registration of Veterinary Medicinal Products (VICH GL49, 2015) [27] was taken as reference to the method validation, which provides a general description of the criteria suitable for the validation of analytical methods used in veterinary drug residue depletion studies. The evaluated performance characteristics were selectivity, matrix effect, the limit of detection (LOD), the limit of quantification (LOQ), linearity, precision, and accuracy. The statistical tests were performed considering the significance level of 0.05.

### 2.6. Medicated Feed

For fish medication, SDM and OMP were incorporated into the commercial fish feed with 28% protein (Pirá Acabamento purchased from Guabi^®^, São Paulo, Brazil). The incorporation process was carried out using gelatin coating, according to recommendations provided by the manufacturer of the commercial drug Romet-30^®^ (Hoffmann-La Roche Inc., Nutley, NJ, USA), as well as previous studies [28]. The target concentrations of SDM and OMP in the medicated feed were 4.17 mg g^−1^ and 0.83 mg g^−1^, respectively. 

Gelatin solution (5%, *w*/*w*) was prepared by dissolving unflavored gelatin in purified water under constant stirring on a hot plate at 50 °C. For every 1 mL of gelatin solution, 64 mg of the drugs (in a 5:1 ratio of SDM:OMP) were then added under stirring. Subsequently, a volume of 25 mL of the warm suspension containing SDM and OMP was added to 250 g of feed under stirring for 5 min. The medicated feed was kept at rest for 2 h, with manual agitation every 10 min, to evaporate the solution. After drug incorporation, the SDM and OMP concentrations in the feed and their homogeneity among feed pellets were evaluated to ensure that the ideal proportion was maintained.

Drug leaching from medicated feed into water during fish medication could compromise the intended dose administration [28]. Therefore, the leaching of SDM and OMP was verified closely following previous studies [29,30,31]. Six pellets of medicated feed (~1 g) were submerged in a beaker containing 900 mL of purified water. The beakers were placed in a Dubnoff metabolic shaker incubator under mild agitation. The pH and temperature were monitored and controlled at 7.1 (±0.6) and 26 (±1.5) °C, respectively. After immersion of the medicated feed into the water, aliquots of 10 mL (*n* = 3) were taken at 5, 10, and 15 min. Immediately after each aliquot was collected, 10 mL of water were added to each beaker to keep the total volume of 900 mL until the next collection. The IS was then added to each aliquot, filtered into a vial, and 5 µL were injected into the LC-MS/MS system. This experiment was performed in triplicate. Simultaneously with the leaching study, the floating capacity of the medicated feed was evaluated.

It is important to highlight that the solubility of SDM and OMP in water are 343 and 1540 μg mL^−1^, respectively [32]. Considering that the maximum concentration of SDM and OMP to be leached into the water would be 3.75 and 0.75 μg mL^−1^, respectively, in 900 mL of water, the solubility of the drug would not compromise its quantitation.

### 2.7. Animals and Experimental Design

The animal experiment was conducted to investigate the concentration of SDM and OMP residues after treatment. The experimental protocol (protocol number 003723/23) was approved by the Ethics Committee for Animal Experimentation, Ethics Committee on the Use of Animals (CEUA) of the School of Agricultural and Veterinary Sciences of UNESP, Jaboticabal, SP, Brazil. The design was performed according to VICH GL48 (2015) [33], a guideline for studies that evaluate the metabolism and kinetics of veterinary drugs in food-producing animals, as well as the VICH GL57 (2019) guideline [34] covering studies on residue depletion in aquatic animals.

One hundred healthy Nile tilapia (0.289 kg average weight) were obtained from a commercial fish farmer and stocked into three plastic tanks (0.31 m^3^, with constant aeration and open water flow). These underwent a 30-day acclimatization period feeding on a commercial diet (Pirá Acabamento, Brazil, with 28% crude protein) fed twice daily at 2% body weight (BW). Fish were provided feed containing SDM and OMP in a 5:1 ratio for five consecutive days (once a day) at a daily dose of 50 mg kg^−1^ BW.

After treatment, fish were sampled and euthanized after 18, 26, 42, 48, 50, 66, 90, 114, and 138 h. On each occasion, ten fish were randomly sampled from tanks and euthanized using a benzocaine immersion bath at 100 mg L^−1^. Muscle plus skin in natural proportions was collected, packed in plastic bags, identified, and stored at −20 °C until analysis. After treatment, the fish resumed feeding the nonmedicated commercial feed until the next sampling.

### 2.8. Depletion Evaluation and Withdrawal Period Estimation

All fillet samples were analyzed in three independent replicates, and the mean concentration of the marker residue (sum of SDM and OMP) per fish was used to calculate depletion. Results with concentrations below the LOQ were reported as half the LOQ. When all or the majority of the data obtained at the same slaughter time were below the LOQ or LOD, the time point was excluded. The marker residue concentrations were plotted as a natural logarithm and as a function of time to fit a linear regression of the depletion profile, assuming the elimination phase of the drug from tissues is a one-compartment model. To evaluate the data obtained from the depletion study and to estimate the withdrawal period, the Excel-workbook-based tool for MRL evaluation was used, developed in 2003 by the Joint Food and Agriculture Organization (FAO)/World Health Organization (WHO) Expert Committee on Food Additives (JECFA). The withdrawal period was estimated as the time when the 99% upper one-sided tolerance limit was below the MRL, at a 95% confidence level, as recommended by USFDA.

## 3. Results and Discussion

### 3.1. Analytical Method Validation

The reliability of the analytical methods developed to quantify SDM and OMP in feed and fish fillets was assessed and the detailed results are presented in Appendix A, respectively. The method involved chromatographic separation commonly used in residue testing laboratories [19]. In the analytical method for SDM and OMP quantification in feed, the r values were higher than 0.99, the coefficient of variance (CV) was lower than 2.4% for intraday and interday precision, and for trueness, the recovery values were between 98.7% and 102.9% (Appendix A). The proposed method for the determination of the drugs in fish fillets involved a modified QuEChERS for sample preparation and LC-MS/MS for quantification. Validation assessment resulted in satisfactory linearity (r > 0.99), intraday and interday precision (CV 0.9–8.4%), and trueness (recovery 85–107%) (Appendix A). The LOQ value was 20 µg kg^−1^ for both analytes. The quantification of SDM and OMP was performed using matrix-matched analytical curves to address the inherent matrix effect. The satisfactory results demonstrated that the analytical methods were fit for the purpose according to Brazilian and international guidelines [26,27].

### 3.2. Drug Feed Incorporation

The oral route for medicated feed is the most convenient form of drug administration in aquaculture systems [35]. Drug incorporation in the feed is a challenging step, especially for SDM and OMP, which differ in their physicochemical properties. Synergistic ratios (SDM:OMP, 5:1) must be maintained in the feed pellets to reduce the risk of under- or overdosing; they also promote better therapeutic outcomes. Since the drugs are water soluble (OMP is more than SDM), the incorporation method must ensure that the drugs are retained in the feed pellets, without leaching, which may otherwise lead to under- or overdosing. A possible strategy to avoid drug leaching is the use of feed coated with polymers [31]. 

In this study, the incorporation of SDM and OMP in the feed involved use of gelatin solution (5%, *w*/*w*) according to the recommendations by the manufacturer of the commercial drug [28]. Gelatin is suitable for coating fish feed because it is a cheap, easily accessible, colorless, tasteless, and water-soluble protein option [36]. Furthermore, because SDM is bitter, suspension of the drugs in gelatin may improve the palatability of the feed [12].

After drug incorporation, the SDM and OMP concentrations in the feed pellets of the same batch were evaluated using the established analytical method. The results (CV < 11%, *n* = 3) indicated uniformity of drugs incorporated in the feed, and that the recommended SDM and OMP ratio was maintained. The incorporation procedure did not affect the feed-floating capacity with the feed pellets remaining on the water surface for up to 15 min. 

One concern with oral route treatment is the potential of the drug to leach from the feed into the aquatic environment [37]. Environmental contamination by these substances may contribute to the proliferation of antimicrobial resistance, one of the main challenges that humans face today [2]. Thus, the procedure for incorporating antimicrobials into feed for aquaculture production has become very critical. In this study, the leaching experiments showed that negligible concentrations of SDM and OMP (<LOQ) were released from the feed into the water for up to 15 min. The incorporation procedure thus retained the drugs in the feed so that it is not readily released into the environment. It is estimated that an adequate period for feed to be available in water until consumption by fish is up to 10 min [38].

### 3.3. Depletion Study and Withdrawal Period Estimation

The weight of the treated fish ranged from 163 to 621 g, with an average of 289 g. Following treatment, it was possible to detect SDM and OMP. Table 1 presents the SDM and OMP concentration determined in Nile tilapia fillets.

These data were evaluated to estimate a residue depletion profile for SDM and OMP in Nile tilapia. First-order decay was presented through the nonlinear mathematical model that better described the elimination kinetics. The terminal elimination of the SDM and OMP and its sum from fish fillets (Figure 1) was considered in the depletion study in order to estimate the withdrawal period for the drugs.

For the withdrawal estimation, data on the concentration of the sum of SDM and OMP were transformed into a logarithmic function through linear regression to estimate the time needed for the residue marker concentration to reach the reference concentration (MRL = 100 µg kg^−1^). When all or most of the reported data from a slaughter time were lower than the LOQ (20 µg kg^−1^), the whole time point was excluded from the analysis. Most of the results of the samples from 168 h after treatment were below the LOQ, and as such, these were not considered for the withdrawal estimation.

The validity of the linear assumption was assessed by checking the homogeneity of variances (Cochran’s test, *p* < 0.05); the lack of it for the linear regression model (ANOVA test, *p* < 0.05); and the normality of errors by plotting the ordered residuals versus their cumulative frequency distribution on a normal probability scale (Appendix A). The obtained graphic representation of the linear regression is presented in Figure 2. 

As seen in Figure 2, the minimum withdrawal period estimated for the residue concentration of OMP plus SDM and below the MRL, considering the 99% percentile at a 95% confidence, was 193 h. This corresponds to 9 days (rounded up from 8.04 days). Since the average water temperature during the medication time was 28 °C, the minimum withdrawal period estimated can be presented as 252-degree days. It is important to note that, since the corresponding straight line does not cross the line corresponding to the MRL, an extrapolation of the results was necessary to estimate the withdrawal period, which may infer statistical errors. 

The USFDA established a 3-day withdrawal period for catfish treated with Romet^®^ (Hoffmann-La Roche Inc., Nutley, NJ, USA) (SDM and OMP present in a ratio of 5:1) at 50 mg/kg for five consecutive days [23]. The Canadian government recommends a minimum withdrawal period of 42 days for fish treated with SDM and OMP (in a ratio of 5:1) when the water temperature is 10 °C or above [15], which represents a minimum of 420-degree days.

Milner et al. (1994) [39] investigated the elimination profile of SDM and OMP after treating Channel catfish (*Ictalurus punctatus*) through medicated feed containing SDM and OMP at a concentration of 50 mg kg^−1^ for five consecutive days at an average water temperature of 25 °C. The data obtained support the present 3-day withdrawal period (equal to 75-degree days) recommended for these antimicrobials considering the MRL of 100 µg kg^−1^. Another study with the same fish species evaluated the antimicrobial residues after oral administration through medicated feed containing 12.5, 25, and 50 mg kg^−1^ BW of SDM and OMP daily, for seven consecutive days (two days of treatment longer than out study) at 28 °C [13]. No SDM or OMP residue was detected in the catfish two days or longer after the last feeding with medicated feed [13].

Samuelsen and Wennevik (2008) [40] evaluated the absorption, tissue distribution, metabolism, and excretion of OMP and SDM in Atlantic salmon (*Salmo salar*) after oral administration of a single dose (30 mg kg^−1^) at an average water temperature of 10 °C. It was observed that by eight days (equal to 80 degree-days) post-treatment, the SDM and OMP residues were below MRL (100 µg kg^−1^).

A previous study compared the elimination kinetics of SDM and OMP in three fish species (Nile tilapia *Oreochromis niloticus*, summer flounder *Paralichthys dentatus*, and walleyes *Sander vitreus*) treated with medicated feed at a daily dose of 50 mg kg^−1^ BW for ten days [41]. The experiments were conducted at the ideal water temperature for each species (30 °C for tilapia, 20 °C for summer flounder and 25 °C for walleyes) and at 5 °C lower than the ideal temperature. The elimination residues of OMP and SDM were separately assessed. According to the authors, no OMP or SDM residues were detected in samples of the edible portion of walleyes (muscle plus skin) collected at day 10 post-treatment or thereafter; for summer flounder, only one fish had a detectable concentration of either residue on day 21 or thereafter, while the residue elimination by Nile tilapia was extremely rapid [41].

Thus, considering the data from the current study and the scientific literature, the need to evaluate the kinetics of drug elimination in different fish species is evident. Factors such as cultivation climatic conditions, dose, treatment period, and medication route all influence drug kinetics [42] and require investigation. For instance, because fish are heterothermic animals, drug metabolism in different fish can be seriously affected by temperature [43]. It is therefore necessary to conduct experiments under local aquaculture conditions and in the countries or regions where drugs to be regulated should be evaluated. To the best of our knowledge, this is the first study that evaluated the residual depletion profile and withdrawal estimation of SDM and OMP in Nile tilapia cultured under Brazilian climatic conditions. Therefore, this study may provide data needed to support the use of SDM and OMP in fish farming systems.

## 4. Conclusions

In this study, two analytical methods were validated and successfully applied for determining OMP and SDM in medicated feed and Nile tilapia fillets. Although SDM and OMP have different physicochemical properties, in this study, the procedure for incorporating these antimicrobials in the feed maintained the recommended ratio for the drugs (SDM and OMP in a 5:1 ratio) and ensured homogeneity of the drug concentrations among the feed pellets. This helped reduce the risk of under- or overdosing and promoted better therapeutic outcomes. Our formulation also prevented the rapid leaching of the antimicrobials from the feed to the water, reducing potential environmental contamination that may contribute to antimicrobial resistance. A fit-for-purpose confirmatory analytical method involving QuEChERS and LC-MS/MS was successfully used to quantify SDM and OMP residues in Nile tilapia fillet on medicated feed at the dose of 50 mg kg BW^−1^ for five consecutive days with an average water temperature of 28 °C. A withdrawal period of 9 days (or 252 °C days) was estimated for eliminating the sum of SDM and OMP residues at concentration levels below the MRL of 100 µg kg^−1^. The findings in this study can provide insights into the regulation of the studied antimicrobials in countries such as Brazil where their use is not approved, but where the lack of regulated drugs for aquaculture is a concern. The information is pertinent to food safety, environmental risks, and human health.

## Figures and Tables

**Figure 1 animals-13-02499-f001:**
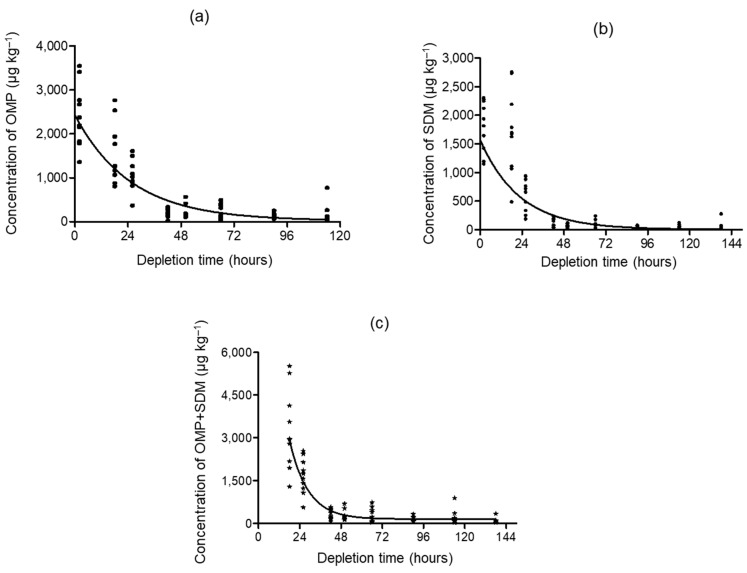
Depletion curves for (**a**) OMP, (**b**) SDM, and (**c**) the sum of OMP and SDM concentrations in Nile tilapia fillet as a function of hours. * = sum of OMP and SDM.

**Figure 2 animals-13-02499-f002:**
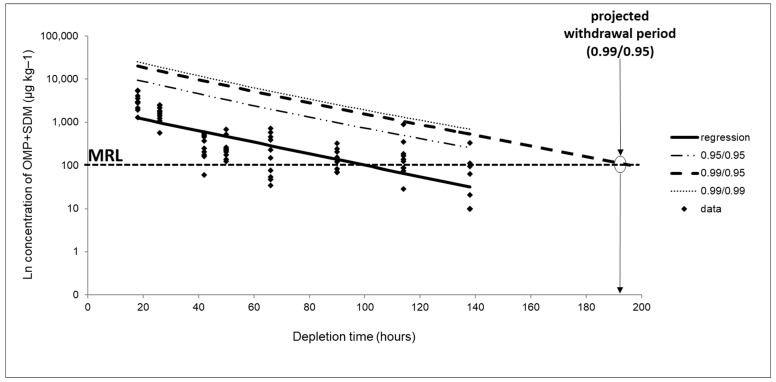
Linear regression graph of the residual depletion curve considering the sum of OMP and SDM residuals in Nile tilapia fillets. Regression: straight line obtained from linear regression of the residual depletion curve. 0.95/95: straight line obtained with 95% confidence considering the 95% tolerance limit. 0.99/95: straight line obtained with 95% confidence considering the tolerance limit of 99%. 0.99/0.99: straight line obtained with 99% confidence considering the tolerance limit of 99%. MRL: Maximum Residue Limit (100 µg kg^−1^).

**Table 1 animals-13-02499-t001:** Concentrations of SDM and OMP in Nile tilapia samples from the depletion study.

Sampling Time (h)	Range Concentration (µg kg^−1^)		Mean Concentration (µg kg^−1^, *n* = 10)		
	OMP	SDM	OMP	SDM	Sum
18	808–2765	485–2756	1545	1714	3259
26	373–1607	189–940	1042	616	1658
42	23–336	37–238	202	130	331
50	110–564	16–491	218	65	283
66	35–121	10–241	207	75	281
90	69–250	10–78	121	37	159
114	29–771	10–124	176	42	218
138	<LOQ–59	<LOQ–278	<LOQ	<LOQ	<LOQ

SDM = sulfadimethoxine; OMP = ormetoprim.

## Data Availability

The datasets generated during and/or analyzed during the current study are available from the corresponding author upon reasonable request.

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
