# Peer review of "Residue Depletion Profile and Estimation of Withdrawal Period for Sulfadimethoxine and Ormetoprim in Edible Tissues of Nile Tilapia (Oreochromis sp.) on Medicated Feed"

_animals, 2023, doi:10.3390/ani13152499_

Round 1

Reviewer 1 Report

Comments to the Author

General comments:

This study investigated Residue depletion profile and estimation of withdrawal period

for sulfadimethoxine and ormetoprim in edible tissues of Nile tilapia (Oreochromis sp.) on medicated feed. The results showed that A withdrawal period of 9 days (or 252 ºC-days) was estimated for the sum of SDM and OMP residues at concentration levels below the maximum residue level of 100 µg kg-1. 

The manuscript could be considered for publication after being major revisions. Some information needs to be provided as follows. 

Specific comments:

1. In Line 217 and Table 1.

-- In Line 217, authors mentioned that fish were sampled and euthanized after 4, 8, 24, 32, 48, 56, 72, 80, 96, 217 and 168 h. However, in Table 1, the sampling time is lined at 2, 18, 26, 42, 50, 66, 90, 114, 138 and 168 h. Please check it again.

2. In the Materials and Methods, statistical methods should be used to analyze the data.

3. Formatting correction

-- Line 202: 1.1. Animals and experimental

-- Line 223: 1.1. Depletion evaluation and withdrawal period estimation

The numbering of the headings must correspond to the context.

4. In Line 238, 3. Results should be rewritten into Results and Discussion

Author Response

Thank you for all your attention and contribution to our manuscript. We considered all your appointments in the revised version of our manuscript.

Follows, point by point, the author's answers to each query raised by the reviewer.

  1. Reviewer: In Line 217, authors mentioned that fish were sampled and euthanized after 4, 8, 24, 32, 48, 56, 72, 80, 96, 217 and 168 h. However, in Table 1, the sampling time is lined at 2, 18, 26, 42, 50, 66, 90, 114, 138 and 168 h. Please check it again.
  • Authors: Thank you. Corrections were made and highlighted in the revised version.

  1. Reviewer: In the Materials and Methods, statistical methods should be used to analyze the data.
  • Authors: We clarify that as mentioned in the manuscript, all obtained data were analyzed according to the reference guide (including the statistical analyzes).

  1. Reviewer: Formatting correction: (1) Line 202: 1.1. Animals and experimental; (2) Line 223: 1.1. Depletion evaluation and withdrawal period estimation; (3) The numbering of the headings must correspond to the context.
  • Authors: Thank you. Corrections were made and highlighted in the revised version.

  1. Reviewer: In Line 238, 3. Results should be rewritten into Results and Discussion
  • Authors: Thank you. Corrections were made and highlighted in the revised version.

Reviewer 2 Report

Line 281: Though the authors described the result of the leaching experiments qualitatively, there is no data or figures shown. Since the authors explained that the contamination of the drugs to the environment is crucial in summary and introduction, this leaching experiment should be exhibited in the paper. 

Line 318, Figure 1: please add the actual equations for the each regression curve in the legend. Then, from the equations, it should be possible to compare the time constant of decay in OMP and SDM. Which drug has longer time constant and could it be discussed in chemical or biological context?

Page 10: This page doesn’t show the numbering of lines.

The contents in this page should be categorized as “4. Discussions”. And the next page content would be titled as “5. Conclusions”.

Line 57. A period is missing in the middle.

Line 76. Please spell out MRLs. Maximum Residue Limits?

Line 202 & 223:  Numbering for the titles is wrong.

Author Response

Thank you for all your attention and contribution to our manuscript. We considered all your appointments in the revised version of our manuscript.

Follows, point by point, the author's answers to each query raised by the reviewer.

  1. Line 281: Though the authors described the result of the leaching experiments qualitatively, there is no data or figures shown. Since the authors explained that the contamination of the drugs to the environment is crucial in summary and introduction, this leaching experiment should be exhibited in the paper.
  • Authors: we clarify that as mentioned in the manuscript, “the leaching experiments showed that negligible concentrations of SDM and OMP (<LOQ) were released from the feed into the water for up to 15 min”. So, there is no accurate data to be presented in this way.
  1. Line 318, Figure 1: please add the actual equations for the each regression curve in the legend. Then, from the equations, it should be possible to compare the time constant of decay in OMP and SDM. Which drug has longer time constant and could it be discussed in chemical or biological context?
  • Authors: We appreciate your comments and understand your point of view. However, we decide to keep the data discussion into our knowledge border, focused on the goal of the study which was evaluate the residual depletion profile aiming to estimate a withdrawal period under food safety issues.
  1. Page 10: This page doesn’t show the numbering of lines. The contents in this page should be categorized as “4. Discussions”. And the next page content would be titled as “5. Conclusions”.
  • Authors: Thank you. Corrections were made and highlighted in the revised version.
  1. Line 57. A period is missing in the middle.
  • Authors: Thank you. Corrections were made and highlighted in the revised version.

Round 2

Reviewer 1 Report

1. In line 44, US$12.342 billion should be revised to  US$ 12.342 billion.

2. In line 102, 98 % should be revised to  98%, and the same as below.

3. In line 161, 4° C should be revised to  4 °C.

4th degree

Author Response

Again, thank you very much for all your attention and contribution. We considered all your comments in the revised version of our manuscript. All of them were highlighted in yellow.

Reviewer 2 Report

The description of the methods are clearer and all the suggestions were implemented well. This paper would contribute to the increase in the safety of aquaculture in Brazil, and other countries.

Author Response

Again, thank you very much for all your attention.